# Architecture for Enhancing Communication Security with RBAC IoT Protocol-Based Microgrids

**DOI:** 10.3390/s24186000

**Published:** 2024-09-16

**Authors:** SooHyun Shin, MyungJoo Park, TaeWan Kim, HyoSik Yang

**Affiliations:** 1Department of Computer Science and Engineering, Sejong University, 209, Neungdong-ro, Gwangjin-gu, Seoul 05006, Republic of Korea; schz307@naver.com (S.S.); taesachi@gabia.com (M.P.); 2Department of Electrical Engineering, Myongji University, Yongin 17058, Republic of Korea

**Keywords:** internet of things, security, microgrid, role-based access control, attribute-based access control, data distribution service

## Abstract

In traditional power grids, the unidirectional flow of energy and information has led to a decrease in efficiency. To address this issue, the concept of microgrids with bidirectional flow and independent power sources has been introduced. The components of a microgrid utilize various IoT protocols such as OPC-UA, MQTT, and DDS to implement bidirectional communication, enabling seamless network communication among different elements within the microgrid. Technological innovation, however, has simultaneously given rise to security issues in the communication system of microgrids. The use of IoT protocols creates vulnerabilities that malicious hackers may exploit to eavesdrop on data or attempt unauthorized control of microgrid devices. Therefore, monitoring and controlling security vulnerabilities is essential to prevent intrusion threats and enhance cyber resilience in the stable and efficient operation of microgrid systems. In this study, we propose an RBAC-based security approach on top of DDS protocols in microgrid systems. The proposed approach allocates roles to users or devices and grants various permissions for access control. DDS subscribers request access to topics and publishers request access to evaluations from the role repository using XACML. The overall implementation model is designed for the publisher to receive XACML transmitted from the repository and perform policy decision making and enforcement. By applying these methods, security vulnerabilities in communication between IoT devices can be reduced, and cyber resilience can be enhanced.

## 1. Introduction

The traditional power grid has a unidirectional structure, where electricity generated at central power plants is transmitted and distributed to consumers. This centralized power grid structure leads to various energy inefficiency issues, such as power loss during transmission and challenges in managing surplus power. To address these problems, extensive research has been conducted in the field of smart grids. Microgrids, as one of the outcomes, feature independent power sources and enable energy self-sufficiency through bidirectional energy and information flow.

To support bidirectional communication in microgrids, communication technologies have been adapted in many areas. Advancements in various IoT (Internet of Things) communication protocols, such as OPC-UA (Open Platform Communications-Unified Architecture), MQTT (Message Queueing Telemetry Transport), and DDS (Data Distribution Service), have enabled bidirectional communication among various components within microgrids. The core objective of IoT communication is to facilitate network connectivity among IoT devices based on various types and purposes of data exchange. This allows seamless data exchange among IoT devices, including sensors and actuators, within microgrids, enabling users to easily access the required data. In other words, microgrids heavily depend on communication protocols for smooth communication among diverse components [1,2,3,4].

These technological advancements, however, simultaneously give rise to security issues in the communication systems of power grids and information overload problems. Firstly, in terms of security issues, malicious hackers may exploit IoT protocols to eavesdrop on data and attempt unauthorized control. Security threats to power grids can lead to significant national disasters. Recent studies indicate that in the event of a nationwide blackout across 15 U.S. states, over 93 million people would be affected, and the associated costs could reach up to one trillion dollars [5]. Despite the stability and predictability of traditional power grids’ control and communication functions using closed networks, past incidents like the nationwide blackout in Italy, Arizona blackout, and Florida blackout have been triggered by large-scale power grid security issues [6]. In the case of microgrids using local-area communication and separating communication and control functions from power devices, a more serious security compromise than traditional power grids may occur, reducing system reliability and hindering stable power system operation. Secondly, failing to differentiate between necessary and irrelevant data and transmitting data to unauthorized users can lead to information overload, disrupting effective power grid operation. Effectively classifying essential and authorized data from vast amounts of data allows for efficient data processing, optimizing data processing efficiency, and minimizing system resource wastage. Therefore, efficient data authorization is more critical than considering communication performance and application validity. In other words, exploring various methods to effectively classify data, confirm data access permissions, and mitigate security threats is essential.

In this work, we proposed a communication security enhancement using standardized access control methods from the international standard IEC 62351 to address communication security threats and information overload issues in microgrids. We applied the open IoT protocol DDS for communication among microgrid components and implemented an RBAC (role-based access control)-based microgrid security enhancement approach. The proposed algorithm can be applied to any IoT protocol that communicates by means of topic with various access control. The proposed algorithm is implemented on top of HTTP (Hyper Text Transfer Protocol) REST (Representational State Transfer) architecture to prove the feasibility and ease of use.

The remainder of this paper will be as follows: Section 2 compares the different access control methods and discusses the recent research in access control schemes. Section 3 discusses the background of this paper. Section 4 proposes the RBAC-based IoT system architecture. Section 5 discusses the details of the proposed system. Section 6 concludes and discusses future works.

## 2. Access Control Methods

A comprehensive review of access control in IoT is summarized in [7]. There are three different categories of access control, i.e., RBAC, ABAC (attribute-based access control), and CBAC (capability-based access control). RBAC is a standardized access control model for managing user permissions in information systems. It has been standardized by the NIST (National Institute of Standards and Technology) and is defined as a method to enhance power system data and communication security in IEC 62351-8 and IEC 62351-90-1 [8,9,10,11]. RBAC is structured around the relationships among roles, users, and permissions. Each role represents specific tasks or responsibilities, and users gain access permissions to particular tasks or functions by being assigned these roles.

Constructing an access control model using RBAC allows for strengthening security by granting users only the minimum necessary permissions. Additionally, managing permissions based on roles facilitates easy adaptation to changes in users’ responsibilities and enables effective permission management. Implementing RBAC in large organizations or complex systems, however, may pose challenges during initial setup or modifications. Managing and maintaining complex relationships among roles, users, and permissions can also be challenging. For example, if an employee holds a managerial role but has different responsibilities and permissions between the headquarters and branches, the managerial role needs to be refined. Furthermore, inadequate monitoring of role or permission changes and assignments can compromise the separation of duties.

ABAC performs access control based on various attributes, providing flexibility for application in diverse environments and allowing high adaptability for adding or modifying new attributes or policies [12,13,14]. Moreover, it offers the advantage of dynamically adjusting access permissions based on changes in states or environments, enhancing security by reflecting the real situations of users. Policy changes or management are simplified, and centralized control can reinforce security.

However, the implementation can be complex due to considerations of various attributes and policies, and it may impact system performance since dynamic access control is performed. Particularly in large-scale systems, considerations for increased load are necessary. Additionally, managing and maintaining various attributes and policies can become challenging as the complexity of policies increases.

Summarizing these characteristics, RBAC is an access control model that grants permissions based on the user’s role. Each role is associated with a specific set of permissions, so when a user assumes a particular role, they acquire the corresponding permissions. ABAC, on the other hand, is a model that controls access considering attributes. This allows for flexible access control but introduces complexities in policy and implementation. In other words, RBAC has a role-centric, straightforward structure, whereas ABAC is a flexible access control model that considers various attributes and conditions.

CBAC uses tokens or keys to grant capabilities for access. It is used in distributed systems where central authority is impractical. The comparison of RBAC and ABAC characteristics has led to various studies. S. Gusmeroli utilized a CL (Capability List) associated with the “subject” instead of using an ACL (Access Control List) [15]. ACL lists roles and permissions for each resource, handling role and permission processing when a specific “object” requests access to the resource. In contrast, CL assigns roles and permissions for individual resources to each “object”, processing access requests for specific resources based on the assigned roles and permissions. ACL contains a list of role and permission information for individual resources, while CL is a list containing role and permission information assigned to resources for each “object”. Traditional RBAC and ABAC implementations commonly adopt ACL. However, ACL introduces complexity as the number of resources or “object” increases, each has different roles and permissions. This complexity arises from the need to reset role and permission information for all “objects” when resources are modified. CL was proposed as a solution to this issue. In CL, each “object” has tailored role and permission information for the required resources, eliminating the need to reset resources even when an additional “object” is added. Therefore, CL is advantageous when scenarios involve an increasing number of resources, evaluating the necessity of resources and resetting role and permission information only for the specific resource. While CL can be beneficial in web services with a continuously increasing number of “objects”, it may be less practical in IoT communication, where devices mostly have fixed roles and permissions for predefined resources.

In the context of IoT implementation within a smart city, the number of “objects” may vary, but nearly all “objects”, such as temperature sensors, AMI (Advanced Metering Infrastructure), sensor-equipped vehicles, and smartphones, have fixed roles and permissions [16]. Consequently, transmitted data are bound to predefined resources, avoiding complexity. Moreover, if adjustments to resources accessed by AMI are required, ACL is advantageous since individual “object” roles and permissions can be processed within the resource. In contrast, using CL would necessitate adjustments for all AMIs based on resource changes. Therefore, in IoT communication, ACL can be considered more advantageous than CL.

Recent studies have focused on security methods applicable to IoT communication using ACL-based approaches, such as RBAC and ABAC. G. Zhang proposed an extended RBAC model to activate RBAC for users and resources in various IoT environments, addressing security issues in IoT systems and introducing new access permissions and security policy features [17]. Q. Liu proposed an RBAC model for multi-domain IoT, integrating hierarchical role structures, new authorization and revocation methods, and access-granting roles facilitating efficient resource sharing. The model demonstrated secure resource sharing, protected user data, and applicability across various domains [18]. S. Ameer proposed a new model utilizing ABAC for secure smart home IoT access control, highlighting the superior flexibility and scalability of ABAC compared to the traditional RBAC model [19]. Furthermore, various ABAC models have been studied to enhance security [20,21,22]. The main advantages of ABAC revealed in these studies are its flexibility and scalability. However, these studies may not adequately address the vulnerabilities and limitations inherent in various communication protocols used in different IoT environments. P. Murugesan adopted ABAC as one of the ACL-based methods applicable to DDS applications [23]. However, this study focuses on access control methods applicable to communication protocols to alleviate concerns about increasing complexity in RBAC. Moreover, in IoT communication, where almost all devices have fixed roles and permissions, there is little potential for a significant increase or segmentation of key roles. Therefore, ABAC cannot be considered the sole alternative to RBAC, and new access control methods integrating the advantages of ABAC, such as NGAC (next-generation access control), are proposed.

Research has been conducted on applying RBAC to communication protocols using publication/subscription methods [24,25,26,27]. However, due to centralization, scalability and flexibility may be limited in these studies. Additionally, these studies target specific scenarios, limiting adaptability to typical publisher/subscriber systems.

While previous studies have primarily focused on static role and permission assignments, the implementation of RBAC using XACML (Extensible Access Control Markup Language) enables flexible access control in dynamic environments. This is particularly relevant in IoT environments like microgrids, where device roles can change depending on the situation. Additionally, the DDS protocol is specialized for the microgrid environment and is defined as the communication protocol in the OpenFMB standard. By integrating DDS and XACML, it is possible to realize fine-grained access control and security management in real-time data distribution systems, which represents a new area not covered by existing research. DDS protocol was used in experiments, but the study ensures applicability to any protocol using XML (Extensible Markup Language) as the data representation format by creating a separate “objects” file.

## 3. Background

### 3.1. XACML (Extensible Access Control Markup Language)

XACML, standardized by OASIS (Organization for the Advancement of Structured Information Standards), is a standard for defining and exchanging XML-based access control policies [28]. It is used in conjunction with RBAC to define and manage robust security policies in various access control scenarios. XACML is employed to extend role-based access control in the RBAC model. While RBAC grants permissions to users based on roles, XACML defines these permissions and allows for centralized management. Therefore, XACML integrates and manages role-based permission definitions, enabling the application of consistent rules in distributed systems. Table 1 shows various types of algorithms that can be defined.

XACML, designed for XML-based policy requests and decision exchange, employs several “Tag” elements. The “Policy” element serves as an introduction to the XACML policy, encompassing namespace declarations, schema URNs (Uniform Resource Names), and the “Policy” instance. The optional “Description” element provides a policy description. The “Rule” element represents the decision regarding the requested “Policy”. The “Target” element specifies the target to which the policy applies, while the “Match” element, a subelement of “Target”, enables the verification of information against matching function values using the “MatchID” attribute value. Multiple “Rules” determine the collection of all “Policies”, and a single “Policy” can consist of multiple “Rules”. A collection of “Policies” can be defined using the “PolicySet” element. Each rule expresses the decision for the requested role through the “Effect” attribute value. XACML incorporates a combination algorithm to process decision outcomes, which is determined by the “RuleCombiningAlgID” attribute of the “Policy” element or the “PolicyCombiningAlgId” attribute of the “PolicySet” element.

In XACML, a PEP signifies the point where access control is directly enforced. When a user or system attempts to access a specific resource, the PEP is invoked to apply policies and perform the role of allowing or denying access. Each PEP listed in Table 2 can be confirmed in Figure 1.

Figure 1 illustrates the basic data flow in XACML in RBAC. The PAP creates policies and policy sets, representing comprehensive policies for specific subjects, and provides them to the PDP. The access requester sends an access request to the PEP. The PEP forwards the access request to its context handler in the native request format, optionally including attributes such as “Subject”, “Resource”, “Action”, “Environment”, and other categories.

The context handler generates the context for XACML requests, optionally adding attributes, and sends it to the PDP. The PDP requests additional attributes like “Subject”, “Resource”, “Action”, “Environment”, etc., from the context handler. The context handler requests attributes from the PIP. The PIP acquires the requested attributes and returns them to the context handler. Optionally, the context handler may include resources in the context during this process. The context handler sends the requested attributes and resources to the PDP. The PDP evaluates the policies and returns the response context, including the decision, to the context handler. The context handler enforces obligations during this process.

XACML has progressed towards standardization using JSON (JavaScript Object Notation) based on REST in HTTP communication. This study uses XML, a document standard used for information exchange in the IEC 61850 standard [29].

### 3.2. DDS (Data Distribution Service)

DDS, a standard middleware for real-time distributed data services, is implemented by the OMG (Object Management Group) and acts as a middleware that facilitates real-time communication between numerous entities [30]. The publisher/subscriber model forms its communication protocol structure, incorporating QoS (Quality of Service) parameters.

The DCPS (Data-Centric Publish–Subscribe) model is adopted by various applications and middleware, including DDS. In contrast to the outdated and inefficient distributed shared memory model lacking scalability and flexibility in networks, the DCPS model is widely embraced in numerous real-time applications.

Figure 2 shows the DCPS structure of DDS applied to the microgrid. In the DCPS structure of the microgrid, the subscriber is the microgrid EMS (Energy Management System), and the publisher may be a sensor, e.g., meter or IED (Intelligent Electronic Device) device. For example, if the subscriber is EMS and the publisher is IED, data exchange is normally possible if the data are in the same domain and topic.

The publisher sends messages using the topic, and the subscriber receives messages from the topic. The publisher publishes messages through the “DataWriter” object, and the message publishing is performed according to the “DataWriter” or the QoS associated with it. The subscriber accesses the received messages through the “DataReader” object and associates data with QoS. The topic, as shown in Figure 2, must have a unique name in the connected area and associates QoS related to the message data type or the data itself. In addition to the QoS of the topic, the QoS of the connected “DataWriter” and publisher controls the publisher’s behavior, and the QoS of the connected “DataReader” and subscriber controls the subscriber’s behavior. In this study, we implemented it using the open-source version provided by OMG.

### 3.3. RBAC (Role-Based Access Control)

IEC 62351 is an international standard specification for security in the IEC TC57 standards, which provides security technologies for IEC 60870, IEC 61850, IEC 61970, IEC 61968, and other standards for information security in smart grid environments using access control and information security methods. IEC 62351-8 provides a model for obtaining access tokens from “subjects to object” in RBAC, such as the “Push” model and “Pull” model [11]. Figure 3 shows the “Push” model and “Pull” model presented in IEC 62351-8.

Figure 3a shows the “Push” model. The “Subject” requests an access token from the identity provider repository, and the identity provider repository delivers an access token containing the “Subject” permission information. The “Subject” that receives the access token attempts to access the “Object” with the access token containing permission information.

Figure 3b shows the “Pull” model. The “Subject” requests access to the “Object”, and the “Object” requests the access token containing the “Subject” permission information from the identity provider repository. The identity provider repository delivers an access token containing the “Subject” permission information to the “Object”, and the “Object” that receives the access token verifies the “Subject” permission based on the confirmed permission and decides the processing for the “Subject”.

The difference between the “Push” model and “Pull” model is whether the “Subject” directly requests the access token from the identity provider repository. Due to this difference, the method in which the “Subject” directly requests/receives its access token and delivers it to the “Object” is named “Push”, and the method in which the “Subject” receives its access token from the “Object” is named “Pull”. In other words, the model in which the access token is related to “Subject to Object” is the “Push” model, and the model in which the access token is related to “Object to Subject” is the “Pull” model. In this study, since the “Object” (publisher) requests the “Subject” (subscriber) permission information through the PDP using XACML when the subscriber requests a topic from the publisher, we will conduct research and implementation using the “Pull” model.

## 4. System Architecture

IEC 61850, a prominent standard in the power field, initially targeted substation standards to ensure interoperability among IEDs within a substation [31]. However, it incurs CPU load and complex protocol stack and is not specialized for microgrids, making it challenging to apply to IoT systems. OpenFMB (Field Message Bus), standardized by NAESB (North America Energy Standardization Board), defines a framework for smart grid standardization in the NIST’s EnergyIoT project within the SGIP (Smart Grid Interoperability Panel). It outlines use cases, PIM (Platform-Independent Models), and communication protocols. Notably, OpenFMB features a structure suitable for microgrid applications, allowing the microgrid access control system designed in this study to refer to the OpenFMB framework [32].

OpenFMB is designed for efficient communication and interaction among devices within a microgrid. It emphasizes supporting real-time data exchange and communication in a distributed environment, adopting a P2P (peer-to-peer) control structure to enable coordination between central controllers and distributed devices. This structure provides greater flexibility and responsiveness compared to a centralized approach. Even if the central controller is far away, OpenFMB can quickly adapt to changes in the grid’s status. The key functions of OpenFMB facilitate seamless communication with various IoT devices within the microgrid, enabling efficient data exchange and collaborative actions among devices. As a result, this enhances the operation and management of the microgrid, contributing to the development of a more stable and efficient power system. The relationship between OpenFMB and DDS lies in OpenFMB using DDS as its data communication protocol. In other words, DDS is employed within the OpenFMB environment to facilitate data communication and interaction among devices. The combination of DDS’s real-time data communication capabilities and OpenFMB’s communication structure tailored for microgrid environments helps improve the efficiency and performance of microgrid systems. Figure 4 illustrates that based on OpenFMB, a microgrid can be defined as a system that establishes a small-scale power network as a locally independent power supply system, meeting energy demand within a specific region. Microgrids typically combine various energy sources, integrating technologies such as solar power, wind power, and the ESS (Energy Storage System).

When applying RBAC to communication within a microgrid, it can be applied to five elements of the microgrid. Firstly, RBAC can be applied to control energy production systems such as generators, solar panels, wind turbines, etc., allowing users or systems with specific roles to set permissions for controlling or monitoring specific energy resources. Secondly, microgrids may include ESS, and RBAC can be used to grant specific roles the authority to efficiently manage and control the ESS. Thirdly, by applying RBAC to the monitoring and management functions of the power network, specific roles can be authorized to check the network status and take necessary actions. Fourthly, RBAC can define the authority for how the microgrid responds and recovers in emergency situations. Specific roles can be given priority or permission to perform specific tasks during emergencies. Finally, RBAC can grant specific roles the authority to perform maintenance and update operations for the microgrid system. By applying RBAC to various elements of communication within the microgrid, each user or system within the microgrid system can operate efficiently with the necessary permissions.

In DDS, when a subscriber requests data for a specific topic from the publisher, it includes its authorization information in the request. Subsequently, the publisher requests access information and authentication details for the requested topic from the repository.

For transmitting the data in JSON format and utilizing the RESTful communication method, the publisher sends the data in JSON format to the repository. In this study, XML was ultimately chosen as the data representation type. Hence, XML was selected as the data representation type for this study. The implementation can be divided into two parts: DDS and XACML. As described in the explanation of RBAC, IEC 62351-8 provides two model options, and this study intends to use the “Pull” model. Accordingly, when applying the DDS and XACML required for the study to Figure 3b, it can be illustrated in Figure 5.

In the “Pull” model, the subject is the subscriber in the DDS, the “Object” is the publisher in the DDS, and the identity provider repository is XACML. The “Subject” requests a publish for a specific topic to the “Object”, and the “Object” uses XACML to request the identity provider repository to determine if the “Subject” has permission to access the specific topic. The identity provider repository evaluates the “Subject” role and permission and sends the evaluation results to the “Object” using XACML, and the “Object” decides whether to “Allow” or “Deny” the topic publish based on the received evaluation results. By applying the “Pull” model in this way, access control based on the roles of IoT devices can be realized. In Figure 5, the “Subject” can be designated as an IoT device; the “Object” can be designated as a CC (Control Center), such as an IoT hub or a network gateway; and the identity provider repository can be designated as the producer of the IoT device. If the producers of IoT devices have different communication methods or representations of communication data, it would be impossible to attempt threat response through RBAC, so standardized standards are necessary. Therefore, in this study, RESTful and HTTPS (Hyper Text Transfer Protocol Secure), as well as XML representation, were selected. XACML and DDS have detailed elements, and these elements provide necessary functions by organically interacting with each other.

Figure 6 shows the placement of XACML and DDS elements in the simplified model established in this study and shows the data flow and sequence for each element. The details on the sequence of the flow can be found in Table 3.

## 5. Proposed System

In this research, we implemented PDP, PIP, and PEP using AuthzForce as the XACML server, separated from a DDS. AuthzForce is an open-source access control server that implements standards, allowing effective management and control of permissions for users or applications. Additionally, we developed functionalities for adding and deleting domains to align with the process of adding and removing devices and subscriber authentication information in IoT communication. To request the addition of a domain, we implemented an XML-based representation of domain information and utilized cURL to make a “Post” request to AuthzForce, following the REST protocol. As a result, unauthorized IoT devices (subscribers) are prevented from receiving data even if they request topics.

Communication between DDS and XACML is achieved through HTTPS REST. The XACML server uses XML as the default format for all requests and responses. To handle HTTPS REST implementation in DDS and process XACML requests and responses, we utilized the cURL library for communication with the server, along with a library supporting XML representation algorithms. Moreover, we developed an XML parser to structurally analyze and extract necessary data from the received XML in HTTPS REST communication. TinyXML, known for its lightweight and efficient handling of XML data, was chosen due to its simplicity and ease of integration into our system. The TinyXML library provided a straightforward API for parsing and manipulating XML documents, allowing us to easily navigate through the XML structure, retrieve specific elements, and extract attributes and text content. This capability was crucial for correctly interpreting the XML data transmitted between the client and the XACML server. By leveraging TinyXML, we were able to implement a parser that parses the XML input, validates its structure, and extracts the required information for further processing. This facilitated seamless communication with the XACML server, which relies on XML representation for data exchange. As a result, the XML parser enabled efficient and accurate access control decision making by ensuring that the data exchanged adheres to the expected format and contains all necessary information.

The main reason AuthzForce was chosen as the XACML server is because it supports standard-based access control decisions. XACML (Extensible Access Control Markup Language) offers the flexibility to express and evaluate complex and diverse access control policies. AuthzForce implements these XACML standards, is extensible, and is provided as open-source, making it easy for developers to customize and extend as needed. Especially in dynamic and diverse access control requirement environments like IoT, AuthzForce provides an effective solution for managing and enforcing complex policies.

The use of HTTPS REST communication between DDS and XACML is due to security and compatibility reasons. HTTPS is a protocol that uses SSL/TLS protocols to ensure secure data transmission, protecting the confidentiality and integrity of data. REST is based on web standards, providing a simple, lightweight, and interoperable interface. Because of these characteristics, REST is widely supported across various platforms and languages, making it suitable for communication between devices or services in an IoT environment. Therefore, the use of HTTPS REST for communication between Da DS and the XACML server enhances security and ensures compatibility with various systems.

These decisions were made carefully, considering the system’s security, scalability, and compatibility, and they can be seen as strategic choices to effectively meet the access control requirements in IoT environments.

Algorithm 1, presented in a pseudocode format, demonstrates the step-by-step process for adding a domain. The algorithm outlines the necessary actions taken by the system to initiate the domain addition request, communicate with the XACML server, and update the domain information accordingly.

**Algorithm 1.** Algorithm of the proposed pseudocode to create domain for subscriber ID.Function Make XML for Domain(subscriber ID)     return some XML codeFunction Call CURL(Target, Method, Params)     url = protocol + “://” + host + “:” + port + “/authzforce/” + target     if protocol == https then          CURLOPT SSL VERIFYPEER = 0L     if method! = GET then          if method == POST then               CURLOPT POST = 1L          else               CURLOPT CUSTOMREQUEST = method     Curleasyper form(url)     return https resultprocedure ACE TMAIN     params = makeXMLForDomain(“NGN”)     buff = callCURL(“domains”, “POST”, params)     dom = parseXML(buff)     subscribeprocess     close;

The basic DDS publish/subscribe communication operates in a way where the publisher delivers data on the requested topic to the subscriber. This procedure utilizes DCPS, with both the subscriber and the publisher accessing the DCPS information repository. This repository acts as a storage for data pertaining to domains and topics. The DCPS information repository is operated through a called “DCPSInfoRepo”.

As shown in Figure 7, both the publisher and subscriber register topics with the “DCPSInfoRepo”. The order in which the publisher and subscriber register does not affect data transmission. If the publisher registers first, it waits until a subscriber, who registered the same topic, registers the topic with the “DCPSInfoRepo” before delivering data to it. On the other hand, if the subscriber registers first, it waits until a publisher, who registered the same topic, registers the topic with the “DCPSInfoRepo” before receiving the data sent by the publisher to the “DCPSInfoRepo”. Generally, the access control behavior may vary depending on the order of topic registration by the publisher and the subscriber, but the most important aspect in the DDS publish/subscribe communication discussed in this paper is the data exchange between the publisher, the subscriber, and the “DCPSInfoRepo”. In this research, we aim to implement a process where the publisher requests subscriber authorization while minimizing the impact on data transmission between the publisher, subscriber, and the “DCPSInfoRepo”. Additionally, when a subscriber requests data for a topic, it includes the generated domain ID and sends it to the publisher. The publisher then forwards the received domain ID to the XACML server’s PDP to check if the corresponding subscriber has permission to access the topic.

However, in Figure 7, the publisher, based on the domain ID received from XACML, has no means to selectively register data in the “DCPSInfoRepo”. For instance, if subscriber 1 receives an “Allow” decision from the PDP while subscriber #2 receives a “Deny” decision, the publisher cannot selectively register data in the “DCPSInfoRepo” based on this information because the published data is destined for all subscribers with the same topic name. Therefore, to achieve selective data transmission based on a subscriber’s domain ID, a separate algorithm is required.

Figure 8 shows the configuration for the selective delivery of topic data using domain IDs by DDS and XACML, aiming to minimize overload through load balancing. The “participant” area in Figure 8 may appear different, but they are the same participant with the same participant number. First, the first subscriber sends its topic name and its domain ID to the participant it belongs to. Then, the second participant forwards the subscriber’s topic name and domain ID to the publisher. The publisher responds with meaningless null data and, simultaneously, the subscriber enters the “listen” state with its domain ID as the topic name. Next, the third publisher sends the domain ID to XACML and requests PDP results. The fourth step involves XACML transmitting the PDP results back to the publisher. Finally, the fifth publisher invokes the PEP and sends the topic data to the participant with subscriber #1’s domain ID as the topic while simultaneously sending an error notice to the participant with subscriber #2’s domain ID as the topic.

Algorithm 2 shows the pseudocode form of the algorithm for selective data transmission. The “DataReaderQoS” class in DDS provides storage space for user data, and in this space, subscribers store the domain ID. The value attribute of the “DataReaderQoS” user data method is of octet type, so it is converted from int to char array to use the domain ID.

**Algorithm 2.** Algorithm for DDS and XACML data flow using domain ID.Procedure Publisher      Create participant      Create topic ← request topic name      if Subscriber is connect then            get domain ID            sending null data to subscriber      else            standbysubscriberconnect      go to XACML      create topic ← domain ID      if Subscriber is connect then            if XACML policy permit then            send topic data to subscriber            else            send error notice      else           standbysubscriberconnectProcedure Subscriber      Create participant      Create topic ← request topic name with domain ID      if Publisher is connect then            set domain ID            receive null data      else            standbypublisherconnect      Create topic ← domain ID      if Publisher is connect then            if XACML policy permits then                  receive topic data from publisher            else                  handle error (e.g., notify user, log error)      else            standbypublisherconnect

Result 1 shows an authorized subscriber requesting data from the publisher for a known topic while simultaneously transmitting a domain ID, “5k71uvbeEemLOQgAJx7RqA”, as user data. The subscriber requests the topic using the domain ID, which is the second topic name in the participant created by the subscriber, and receives the requested data from the publisher.

**Result 1.** Result message from authorized subscriber.--------------------------------------------------------------------------------------------[Microgrid Resource List][5k71uvbeEemLOQgAJx7RqA]--------------------------------------------------------------------------------------------Resource.priority = 0Resource.status = 1  resource_type =   resource_id = 0  inspector =   issue_count = 0  status_comment = Resource.priority = 0Resource.status = 1  resource_type = “Solar Panel”  resource_id = 99  inspector = “Technician A”  issue_count = 0  status_comment = “Panel efficiency severely reduced”Resource.priority = 0Resource.status = 1  resource_type = “Solar Panel”  resource_id = 100  inspector = “Technician A”  issue_count = 1  status_comment = “Panel efficiency severely reduced”Resource.priority = 0Resource.status = 1  resource_type = “Solar Panel”  resource_id = 101  inspector = “Technician A”  issue_count = 2  status_comment = “Panel efficiency severely reduced”

In Result 2, it can be observed that when an unauthorized subscriber requests the topic, no data are received. However, after transmitting the domain ID and creating the “5k71uvbeEemLOQgAJx7RqA” topic, the subscriber receives actual published data, confirming its reception.

**Result 2.** Data received in subscriber on PDP decision to permit.--------------------------------------------------------------------------------------------[Microgrid Resource List]--------------------------------------------------------------------------------------------Resource.priority = 0Resource.status = 1  resource_type =   resource_id = 0  inspector =   issue_count = 0  status_comment = --------------------------------------------------------------------------------------------[5k71uvbeEemLOQgAJx7RqA]--------------------------------------------------------------------------------------------Resource.priority = 0Resource.status = 1  resource_type = “Solar Panel”  resource_id = 99  inspector = “Technician A”  issue_count = 0  status_comment = “Panel efficiency severely reduced”Resource.priority = 0Resource.status = 1  resource_type = “Solar Panel”  resource_id = 100  inspector = “Technician A”  issue_count = 1  status_comment = “Panel efficiency severely reduced”Resource.priority = 0Resource.status = 1  resource_type = “Solar Panel”  resource_id = 101  inspector = “Technician A”  issue_count = 2  status_comment = “Panel efficiency severely reduced”                     .                     .Resource.priority = 0Resource.status = 1  resource_type = “Solar Panel”  resource_id = 108  inspector = “Technician A”  issue_count = 9  status_comment = “Panel efficiency severely reduced”Resource.priority = 0Resource.status = 2Resource.priority = 0Resource.status = 2Resource.priority = 0Resource.status = 2--------------------------------------------------------------------------------------------

Furthermore, in Result 3, when the domain ID value is changed to a different one, the subscriber experiences a deny and does not receive any data.

**Result 3.** Data receive in subscriber on PDP decision to deny.--------------------------------------------------------------------------------------------[Microgrid Resource List]--------------------------------------------------------------------------------------------Resource.priority = 0Resource.status = 1  resource_type =   resource_id = 0  inspector =   issue_count = 0  status_comment = --------------------------------------------------------------------------------------------[Fake Domain ID]--------------------------------------------------------------------------------------------Resource.priority = 0Resource.status = 1  resource_type =   resource_id = 0  inspector =   issue_count = 0  status_comment = --------------------------------------------------------------------------------------------

## 6. Conclusions

In this study, we proposed an access control method that leverages RBAC implementation with DDS and XACML. To achieve this, we integrated AuthzForce as the XACML server, located separately from the DDS. To facilitate communication with the XACML server, we harnessed the cURL library to establish secure HTTPS REST connections. Moreover, to enhance flexibility, we developed a separate object file, making it compatible with various applications. One of the key innovations in our approach is the implementation of server address and port modification functionality, allowing the XACML server to be applied as a remote server. This enables remote access control management, which can be particularly useful in IoT environments. For seamless communication between the DDS and XACML components, we developed an XML parser as another separate object file. This XML parser facilitates communication with the XACML server using XML representation, enabling smooth data exchange and access control decision making. By decoupling the functionalities of XACML, cURL, XML, and the publish/subscribe application, we have achieved a highly modularized design. As a result, our access control mechanism is not limited to DDS alone, but it can be easily adapted to other publish/subscribe-based IoT communication protocols such as MQTT and OPC-UA. This versatility broadens the scope of application and encourages wider adoption of the access control framework. With this logical implementation, our approach enables access control based on subscriber roles and attributes, providing an added layer of security. It mitigates unauthorized access attempts, leading to better resource utilization and enhanced information security. Looking ahead, our research roadmap includes conducting in-depth performance analysis for communication between publishers and subscribers in both 1:n and n:m multicast communication scenarios. Understanding the communication dynamics will aid in developing more advanced and efficient algorithms for access control.

Based on these findings, this study contributes in the following ways. It proposes an innovation in the power structure through microgrids, overcoming the limitations of traditional power grids and attempting to enhance efficiency and stability. Additionally, to address the security issues arising from the introduction of microgrids in data communication systems, the study suggests a system utilizing access control to increase the reliability of power networks. These points contribute to shaping the future of power systems, aiding in the secure integration of advanced systems like microgrids to build a more intelligent and sustainable energy infrastructure.

In the future, it will be possible to implement an access control mechanism that operates effectively in various IoT environments while minimizing power consumption and resource usage. As the approach expands with the growing deployment of IoT, it will be sustained sustainably and remain scalable through optimal resource utilization. Moreover, leveraging the proposed communication security enhancement system, it is feasible to establish microgrids that ensure interoperability with existing power grids. This is seen as a crucial agenda contributing to the establishment of a sustainable energy infrastructure.

## Figures and Tables

**Figure 1 sensors-24-06000-f001:**
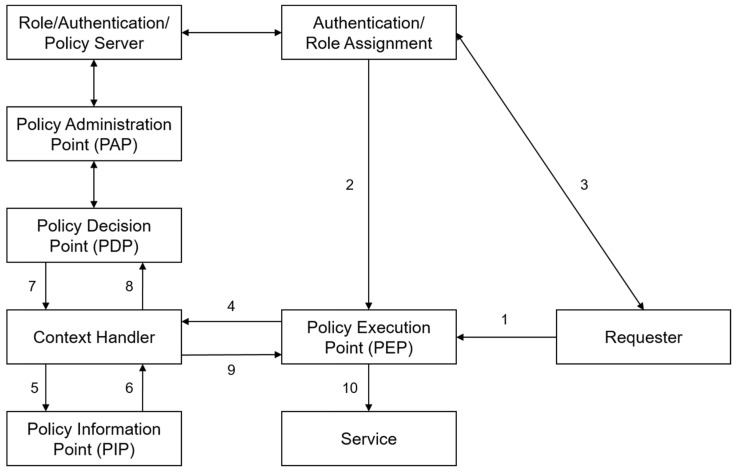
Data flow of XACML.

**Figure 2 sensors-24-06000-f002:**
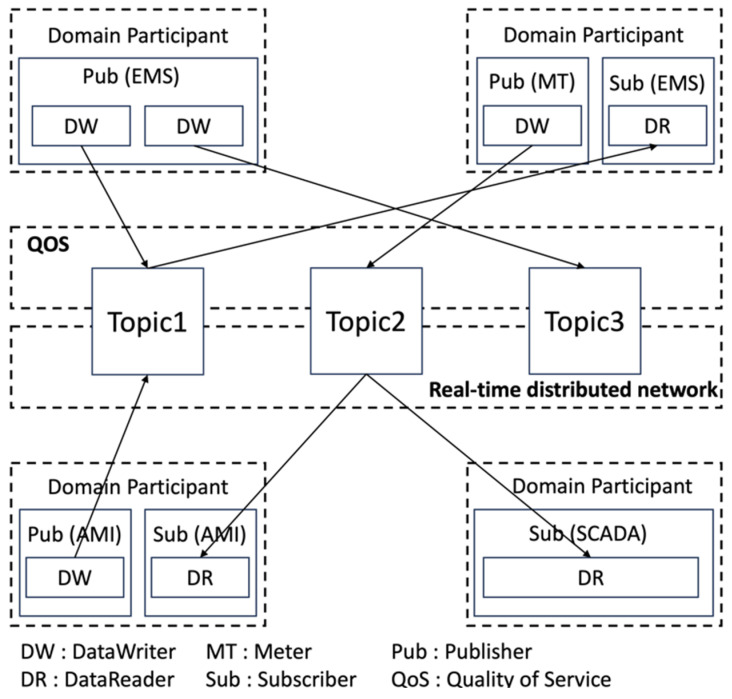
DCPS structure in microgrid.

**Figure 3 sensors-24-06000-f003:**
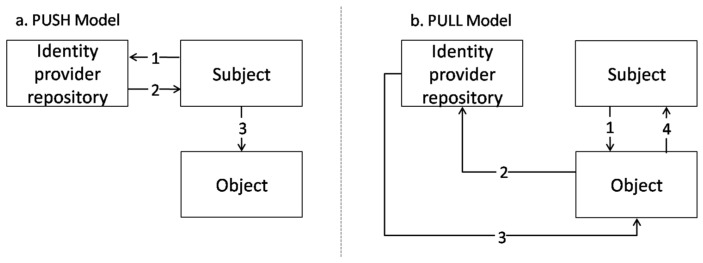
“Push” and “Pull” model in IEC 62351-8 [10].

**Figure 4 sensors-24-06000-f004:**
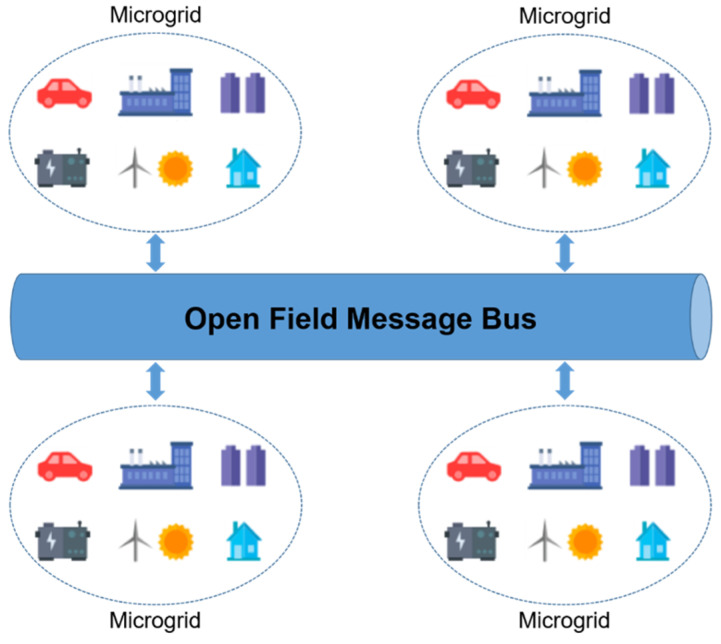
OpenFMB architecture [32].

**Figure 5 sensors-24-06000-f005:**
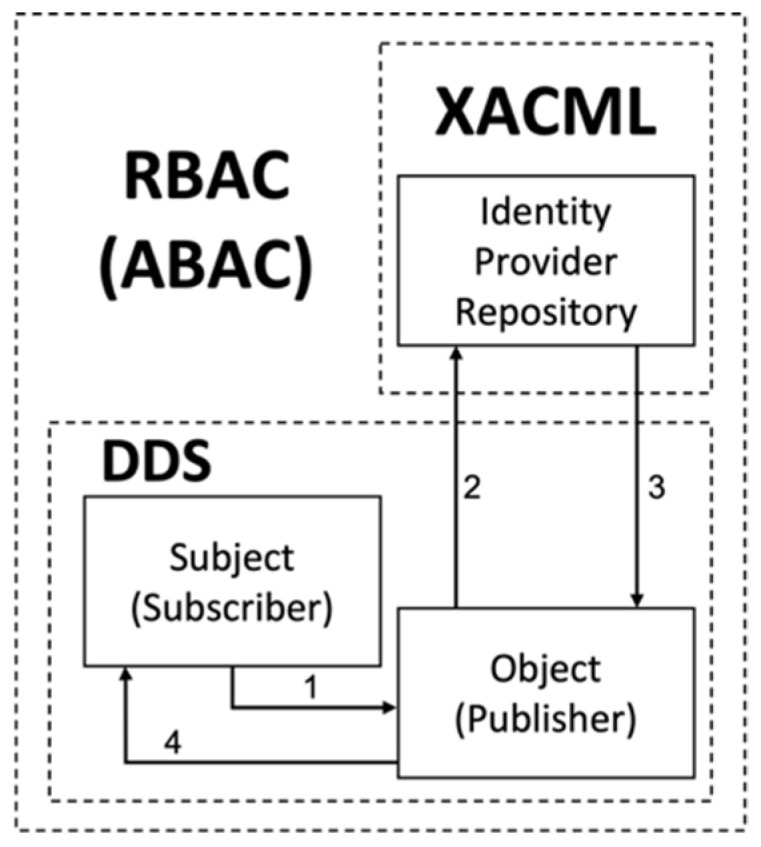
DDS and XACML into the concept of draft idea.

**Figure 6 sensors-24-06000-f006:**
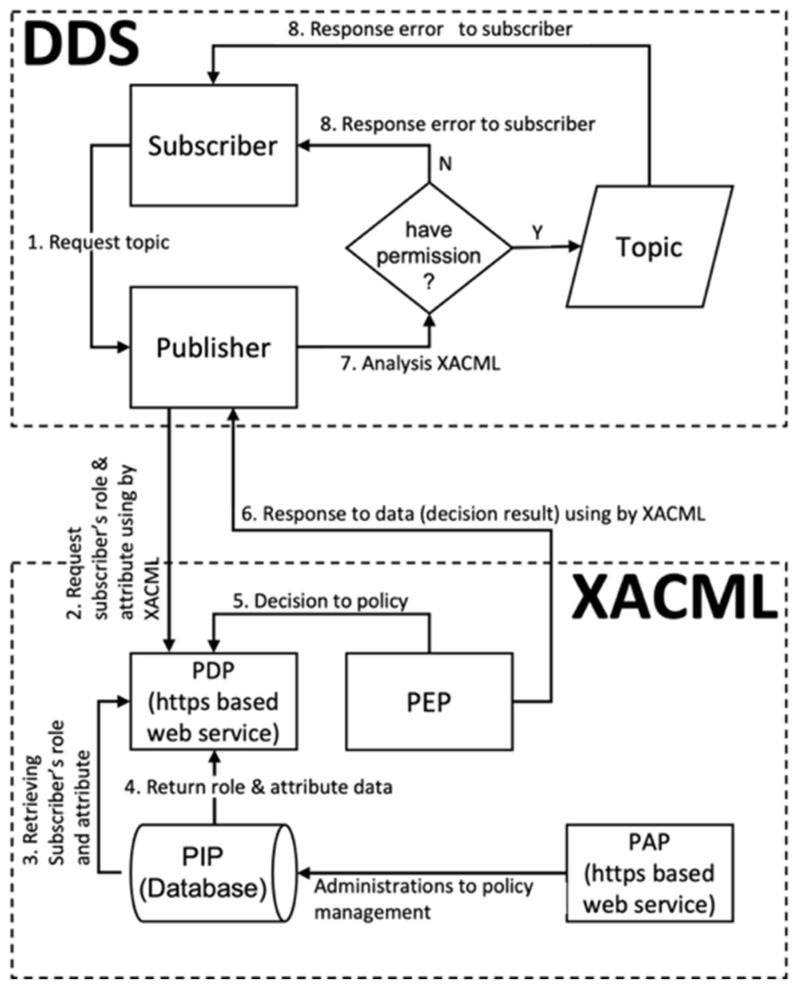
Overall architecture.

**Figure 7 sensors-24-06000-f007:**
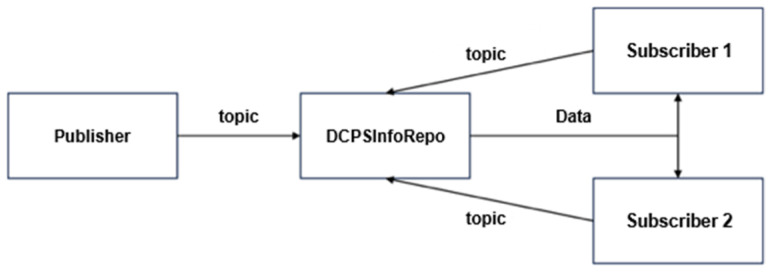
Communication of publish and subscribe on DDS.

**Figure 8 sensors-24-06000-f008:**
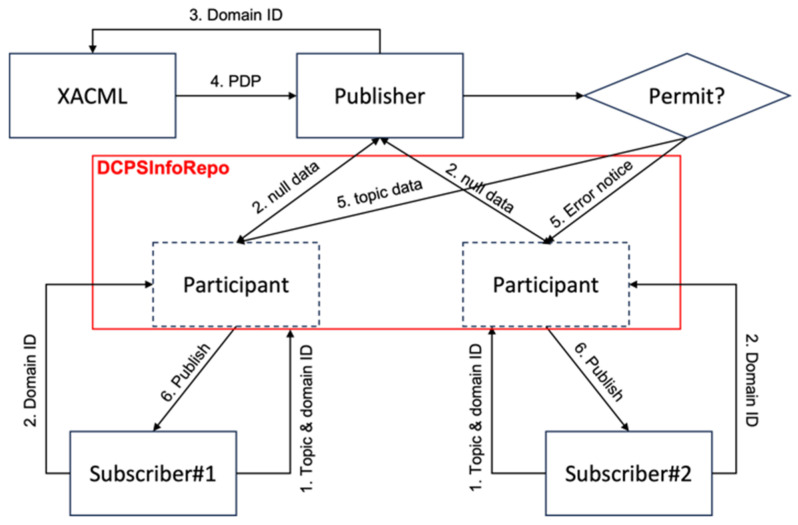
DDS and XACML data flow using domain ID.

**Table 1 sensors-24-06000-t001:** Types of combined algorithms in XACML [28].

Algorithm	Description
Deny-Overrides	Deny First Algorithm
Permit-Overrides	Permit First Algorithm
First-Applicable	First-Applicable Algorithm
Only-One-Applicable	Unique Value Assignment Algorithm

**Table 2 sensors-24-06000-t002:** Terminology of policy point in XACML [28].

Point	Description
PAP (Policy Administration Point)	Access Policy Management
PDP (Policy Decision Point)	Access Policy Evaluation and Decision
PEP (Policy Enforcement Point)	Enforcement of Access Control based on Evaluated and Decided Policies
PIP (Policy Information Point)	Access Policy Storage Created and Managed by PAP

**Table 3 sensors-24-06000-t003:** Explanation of flow diagram of Figure 6.

No	Description
1	The subscriber requests the publisher to publish specific topics, and when making the request, the subscriber also provides its account information.
2	The publisher configures the subscriber’s role and attributes for the topic using XACML and requests the PDP. The communication method uses RESTful based on HTTPS.
3	The PDP queries the subscriber’s role and attributes from the PIP.
4	The subscriber’s role and attributes retrieved from the PIP are returned to the PDP.
5	The PDP evaluates the returned subscriber’s role and attributes and delivers the evaluation result to the PEP.
6	The PEP composes the evaluated result into XACML and sends it to the publisher. Communication uses an HTTPS session that is already established.
7	The publisher analyzes the received XACML and determines whether the subscriber has the authority based on the role or attributes for the relevant topic.
8-Y	If the subscriber has the authority, the publisher publishes the topic data to the subscriber.
8-N	If the subscriber does not have the authority, the publisher publishes “no authority” to the subscriber.
Etc.	The PAP, which is constructed as a web page based on HTTPS, configures the subscriber account’s role and attributes for the topic, and the configured content is stored in the PIP’s database for management.

## Data Availability

The data presented in this study are available on request from the corresponding author.

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
