# Peer review of "Architecture for Enhancing Communication Security with RBAC IoT Protocol-Based Microgrids"

_sensors, 2024, doi:10.3390/s24186000_

Round 1

Reviewer 1 Report

Comments and Suggestions for Authors

In order to solve the problem of reduced efficiency caused by the one-way flow of energy and information in traditional power grids, this paper introduces the concept of microgrid with bidirectional flow and independent power supply. At the same time, in order to solve the security problem of microgrid communication system, an access control method based on RBAC of DDS and XACML is proposed. AuthzForce is integrated as a XACML server separated from DDS, and a secure HTTPS REST connection is established using cURL library. In addition, in order to achieve seamless communication between DDS and XACML components, an XML parser is developed as another separate object file.

There are some issues in the manuscript that need to be revised by the authors before acceptance. The problems in the manuscript are detailed below.

1.       In the introduction, the author should introduce the rest of the paper and briefly explain the arrangement of the chapters.

2.       In Section 3, the author introduced related work but lacked the latest research results. It is recommended to add cutting-edge research and cite the latest literature from 2022 to 2024 to increase the overall value of the article.

3.       The content in Figure 3 is not clear enough. The author is advised to revise it and provide a clearer image to help understand the paper proposal. Other pictures in the paper also need further polishing and improvement.

4.       This paper lacks experimental analysis to prove the effectiveness of the solution.

5.       The conclusion of the paper is a bit too long and should be concise and highlight the contribution of this article. Please further improve it.

6.       The overall structure of the paper needs further adjustment. The content is too redundant. Please reduce unnecessary descriptions. The focus should be on highlighting the contributions and work of this paper.

7.     There are some grammatical errors and inappropriate expressions in the manuscript, which do not conform to English logic and require further optimization.

Comments on the Quality of English Language

minor editing

Author Response

Response to Reviewer 1 Comments

Authors are grateful to the reviewers on their invaluable comment to improving the quality of the paper.

Point 1: In the introduction, the author should introduce the rest of the paper and briefly explain the arrangement of the chapters.

Response 1: The authors add the introduction the arrangement of the chapters

Point 2: In Section 3, the author introduced related work but lacked the latest research results. It is recommended to add cutting-edge research and cite the latest literature from 2022 to 2024 to increase the overall value of the article.

Response 2: The authors add more latest references

Point 3: In Section 3, the author introduced related work but lacked the latest research results. It is recommended to add cutting-edge research and cite the latest literature from 2022 to 2024 to increase the overall value of the article.

Response 3: We change the organization of Figure 3 for clear understanding.

Point 4: In Section 3, the author introduced related work but lacked the latest research results. It is recommended to add cutting-edge research and cite the latest literature from 2022 to 2024 to increase the overall value of the article.

  •  

Response 4: The authors add analysis of experiments

Point 5: The conclusion of the paper is a bit too long and should be concise and highlight the contribution of this article. Please further improve it.

Response 5: The authors revise the conclusion

Point 6: The overall structure of the paper needs further adjustment. The content is too redundant. Please reduce unnecessary descriptions. The focus should be on highlighting the contributions and work of this paper.

Response 6: The authors reorganize the structure of paper

Point 7: There are some grammatical errors and inappropriate expressions in the manuscript, which do not conform to English logic and require further optimization

Response 7: Grammatical errors are corrected by the native English speaker.

Reviewer 2 Report

Comments and Suggestions for Authors

Its a nice work but I have below recommendations for the authors to make the quality better.

- When using any abbreviation, make sure you are explaining or detailing the full form of that abbreviation before using it through out the paper. Example: XACML, ABAC etc.

- Categorized use of ABAC, RBAC CBAC or NGAC need to be discussed in detail, i.e. when and where it is beneficial and why. Please highlight if one can be subset of the other.

- I have some doubts in "Algorithm 2. Algorithm for DDS and XACML Data Flow Using by Domain ID", revisit the Procedure Subscriber flow.

- There is a mention in the paper that 'the authors have built a XLM parser to facilitate data in XACML. Need more insights about the parser in this parser.

Comments on the Quality of English Language

Paper is good but few English issues are there. Please run it through the Grammarly to correct those. 

Author Response

Response to Reviewer 2 Comments

Authors are grateful to the reviewers on their invaluable comment to improving the quality of the paper.

Point 1: When using any abbreviation, make sure you are explaining or detailing the full form of that abbreviation before using it through out the paper. Example: XACML, ABAC etc.

Response 1: We explain the full form of each abbreviation at the first time used

Point 2: Categorized use of ABAC, RBAC CBAC or NGAC need to be discussed in detail, i.e. when and where it is beneficial and why. Please highlight if one can be subset of the other.

Response 2: We revise the related work sections as the reviewer pointed out

Point 3: I have some doubts in "Algorithm 2. Algorithm for DDS and XACML Data Flow Using by Domain ID", revisit the Procedure Subscriber flow.

Response 3: We revise the Algorithm 2.

Point 4: There is a mention in the paper that 'the authors have built a XLM parser to facilitate data in XACML. Need more insights about the parser in this parser.

Response 4: We add sentences for describing the parser.

Reviewer 3 Report

Comments and Suggestions for Authors

The evaluated work demonstrates significant potential by presenting a practical approach to controlling attributes in a communication system, providing an additional layer of security and mitigating unauthorized access in various communication protocols within the research context. However, problems were identified in the presentation of the elements of the academic material used and the research variables.

The introduction can be improved by incorporating more recent references on the subject and further developing the corrections proposed by the authors. It would be useful to include a comparison of these proposals with current practices used in traditional electrical networks. In sections 2 and 3, some paragraphs lack the necessary theoretical foundation to support the authors' claims. Additionally, some minor grammatical errors were observed, indicating the need for a review to correct these mistakes.

Comments on the Quality of English Language

To ensure that the article meets the linguistic and academic standards required by international journals, we recommend having a professional review of the English text. This will help enhance the clarity, flow, and accuracy of the language used and ensure that all technical terms are correct and appropriately applied.

I suggest considering the services specializes in academic and scientific reviews. They offer a detailed analysis of grammar, style, and consistency, ensuring that the manuscript is ready for publication in high-impact journals.

Author Response

Response to Reviewer 3 Comments

Authors are grateful to the reviewers on their invaluable comment to improving the quality of the paper.

Point 1: The introduction can be improved by incorporating more recent references on the subject and further developing the corrections proposed by the authors. It would be useful to include a comparison of these proposals with current practices used in traditional electrical networks. In sections 2 and 3, some paragraphs lack the necessary theoretical foundation to support the authors' claims.

Response 1: We revise the introduction section, add more references, and describes the contribution of proposed architecture.

Point 2: Additionally, some minor grammatical errors were observed, indicating the need for a review to correct these mistakes

Response 2: Grammatical errors are corrected by the native English speaker

Reviewer 4 Report

Comments and Suggestions for Authors

The author proposed an architecture for enhancing communication security in microgrids by using Role-Based Access Control (RBAC) and IoT protocols.

However, the paper's motivation is unclear. The significance of the research problem needs to be clearly articulated and well-motivated. The authors should summarize the main contributions of the paper in the Introduction section.

-The paper requires proofreading, as some acronyms are used without definition.

-Additionally, the authors should summarize the main contributions of the paper.

-The title of Table 1 should be revised (e.g., "Types of Combined Algorithms in XACML [24]").

-Figure 6 needs improvement.

- The authors should compare and evaluate their solution with the state-of-the-art approaches.

Comments on the Quality of English Language

Some phrases in the text need improvement; proofread the paper.

Author Response

Response to Reviewer 4 Comments

Authors are grateful to the reviewers on their invaluable comment to improving the quality of the paper.

Point 1: However, the paper's motivation is unclear. The significance of the research problem needs to be clearly articulated and well-motivated. The authors should summarize the main contributions of the paper in the Introduction section

Response 1: We revise the introduction section and adds contribution of the paper

Point 2: The paper requires proofreading, as some acronyms are used without definition

Response 2: We explain the full form of each abbreviation at the first time used

Point 3: Additionally, the authors should summarize the main contributions of the paper.

Response 3: We revise the introduction section to address the main contribution of the paper.

Point 4: The title of Table 1 should be revised (e.g., "Types of Combined Algorithms in XACML [24]").

Response 4: We change the title of Table 1 as proposed.

Point 5: Figure 6 needs improvement

Response 5: Figure 6 is changed

Point 6: The authors should compare and evaluate their solution with the state-of-the-art approaches

Response 6: The authors should compare and evaluate their solution with the state-of-the-art approaches with additional recent research.
